# SAU-Net: Efficient 3D Spine MRI Segmentation Using Inter-Slice Attention

**Yichi Zhang**[1]
**Lin Yuan**[1,2]
**Yujia Wang**[1]
**Jicong Zhang**[*1,3,4,5]

[1]*School of Biological Science and Medical Engineering, Beihang University, Beijing , China*
[2]*School of Biomedical Engineering, Capital Medical University, Beijing , China*
[3]*Hefei Innovation Research Institute, Beihang University, Hefei, China*
[4]*Bejing Advanced Innovation Centre for Biomedical Engineering, Beihang University, Beijing, China*
[5]*Bejing Advanced Innoration Centre for Big Data-Based Precision Medicine, Beihang University, Beijing, China*

## Abstract

Accurate segmentation of spine Magnetic Resonance Imaging (MRI) is highly demanded in morphological research, quantitative analysis, and diseases identification, such as spinal canal stenosis, disc herniation and degeneration. However, accurate spine segmentation is challenging because of the irregular shape, artifacts and large variability between slices. To alleviate these problems, spatial information is used for more continuous and accurate segmentation such as by 3D convolutional neural networks (CNN) . However, 3D CNN suffers from higher computational cost, memory cost and risk of over-fitting, especially for medical images where the number of labeled data is limited. To address these problems, we apply the attention mechanism for the utilization of inter-slice information in 3D segmentation tasks based on 2D convolutional networks and propose a spatial attention-based densely connected U-Net (SAU-Net), which consists of Dense U-Net for extraction of intra-slice features and an inter-slice attention module (ISA) to utilize inter-slice information from adjacent slices and refine the segmentation results. Experimental results demonstrate the effectiveness of ISA as well as higher accuracy and efficiency of segmentation results of our method compared with other deep learning methods.

**Keywords:** spine segmentation, MRI, deep learning, inter-slice attention

## 1. Introduction

Due to changes in lifestyles caused by social development, spine-related diseases like spinal canal stenosis, disc herniation and degeneration have become common clinical diseases, posing a serious threat to patients' health. Clinically, spine MRI could show the anatomical structure and relative position of vertebra steadily and distinctly, which has become one of the most common and effective methods for diagnosis (Korez et al., 2015).

---

* Corresponding author. E-mail address: jicongzhang@buaa.edu.cn

Accurate and robust segmentation of spine MR images is an essential tool for identification and quantitative analysis of diseased region. Segmentation results can be used to diagnose various characteristic diseases such as disc degeneration (Wu et al., 2014), adolescent idiopathic scoliosis (AIS) (Guerroumi et al., 2019) and Lytic bone metastases in thoracolumbar spine (Yao et al., 2007) after subsequent processing, considerably aid surgical planning for doctors. Besides, the results can also be used to assist in clinical treatment of spine classification, 3D reconstruction of spine and associated research.

However, spine segmentation is still considered a difficult task owing to many challenges such as unclear spine boundaries, abnormal spinal curvature and intricacy of vertebral structures. Traditional segmentation methods which rely on manual intervention and prior knowledge usually require a lot of effort and time. Besides, these methods are prone to errors on account of variability between operators. Automatic image segmentation could increase precision by eliminating the subjectivity and exhaustive processes. Consequently, extensive work is devoted to the research of automatic spine segmentation.

With the success of deep learning in medical imaging, most spine segmentation methods published recently are based on deep learning and have replaced explicit modeling of the vertebral shape. For example, (Sekuboyina et al., 2017) segmented the lumbar vertebrae in 2D sagittal slices using CNN for pixel labeling, then a simple multi-layer perceptron estimated a bounding box to identify the region of interest in the image. (Lessmann et al., 2019) used iterative fully convolutional neural network (FCN) that performs multiple tasks concurrently for spine segmentation.

Although Spine segmentation based on 2D slices has produced superior segmentation results with satisfactory performance, these methods ignore the spatial continuity between slices, limiting further improvements in the segmentation performance. In order to deal with these limitations, segmentation methods based on 3D convolutions were proposed like 3D U-Net (Cicek et al., 2016) and V-Net (Milletari et al., 2016). These methods achieved better performance in 3D medical image segmentation tasks compared with 2D methods. For spine segmentation, with the addition of increased dimensionality, the disadvantages of 3D networks are gradually emerging. Despite various creative 3D network training assisted by the scale-down of the inputs (Chen et al., 2020) the large calculations caused by 3D full-size inputs resulting in high computer memory requirements and unnecessary time costs cannot be fundamentally mitigated, especially for high resolution images.

To deal with these problems, a novel network architecture called spatial attention-based densely connected U-Net (SAU-Net) is proposed for 3D spine MRI segmentation. First, the initial 3D image is decomposed into a stack of 2D slices [1]. Then a Dense U-Net structure is constructed to to get rough probability results based on intra-slice information. In the end, an inter-slice attention module is appended to capture and fuse 3D inter-slice spatial information with 2D contextual information, therefore refines the voxel-wise segmentation results.

In summary, our main contributions can be summarized as follows:

- To the best of our knowledge, we are the first to apply the attention mechanism for the utilization of inter-slice information in 3D segmentation tasks based on 2D convolutional networks and propose inter-slice attention module (ISA) .

---

1. In this paper, slices refers specifically to sagittal slices of 3D spine MR images.

- We propose a novel structure called spatial attention-based densely connected U-Net (SAU-Net) for effective and accurate spine segmentation from 3D MR images.

In the following paragraphs, we firstly introduce some related work about deep-learning-based image segmentation. Then we demonstrate the detail of our proposed method. After that, a series of experiments are conducted to prove the effectiveness and superiority of ISA and SAU-Net.

## 2. Related work

### 2.1. Image Segmentation by Deep Convolutional Neural Networks

Image segmentation aims to understand images in pixel level and label each pixel into a certain class. Deep learning based object detection and semantic segmentation in computer vision has made a big advancement recently. The concept of convolution neural network (CNN) was first introduced by (Lecun et al., 1989) and it was widely applied for classication(Krizhevsky et al., 2012). Fully convolutional networks (FCN) proposed by (Long et al., 2015) is a landmark in image segmentation. It applied CNN to dense prediction and first realized end to end segmentation by replacing fully connected neural layers with convolutional neural layers. With the development of deep learning, more and more studies on image segmentation using neural networks have been proposed. The most popular structure for medical image segmentation is U-Net proposed by (Ronneberger et al., 2015). The architecture of U-Net fused the features of different scales by concatenating the feature maps of the downsampling layers and the corresponding upsampling layers. For segmentation of 3D images, (Cicek et al., 2016) proposed 3D U-Net architecture that inputs a 2D slice sequence of 3D images. (Milletari et al., 2016) evolved U-Net into V-Net by using 3D convolution kernels to extract features from images. These methods have achieved relatively good results, showing the advantages of encoder-decoder structures based on U-Net in medical image segmentation tasks.

### 2.2. Attention-based Methods for Image Segmentation

Attention can be viewed as using information transferred from several subsequent feature maps to select and localize the most discriminative part of the feature maps. (Oktay et al., 2018) proposed attention gates to learn to suppress irrelevant regions while highlighting salient features useful for a specific task. (Hu et al., 2019) proposed a selection mechanism where feature maps are first aggregated using global average pooling and then reduced to a single channel descriptor with an activation gate applied to highlight the most discriminant features. For medical image segmentation, (Sinha and Dolz, 2019) proposed a multi-level attention based architecture for abdominal organ segmentation from MR images. (Qin et al., 2018) proposed a dilated convolution base block to preserve more detailed attention in 3D medical image segmentation. (Sekuboyina et al., 2017) proposed an attention net for spine detection before using 3D U-Net for segmentation.

## 3. Methods

The general architecture of our proposed SAU-Net is shown in Figure1. The input 3D MR image $X \in R^{d \times l \times w}$, where $d, l, w$ represents the depth, length and width of the images respectively. At first, the 3D volume is divided into a sequence of slices of 2D images $[x_1, x_2, ..., x_d] \in R^{l \times w}$. Then, an encoder-decoder Dense U-Net structure is utilized to capture intra-slice information and obtain rough probability results of each slice $[\widehat{y_1}, \widehat{y_2}, ..., \widehat{y_d}]$. In the end, an inter-slice attention module is appended to capture and fuse 3D spatial information with 2D contextual information, therefore refines the voxel-wise segmentation results $\widehat{Y} \in R^{d \times l \times w}$ .

In the following subsections, we will introduce the detailed structure of Dense U-Net and the design of inter-slice attention module.

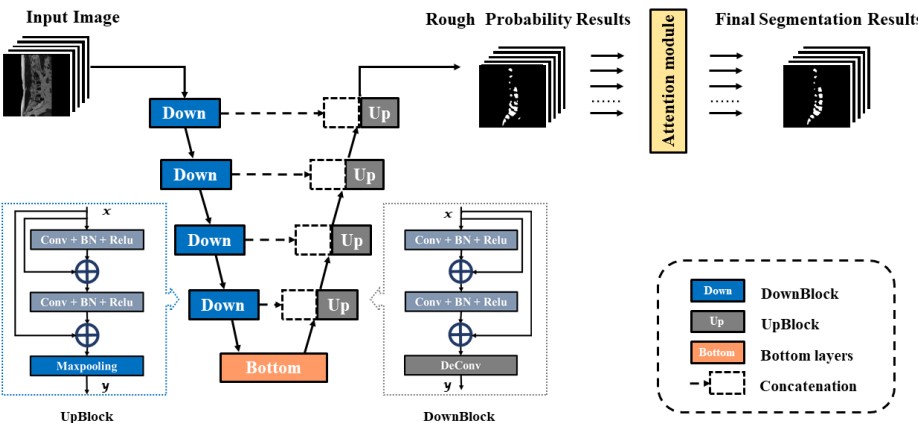

Figure 1: The general architecture of SAU-Net for spine segmentation. At first, input 3D images are predicted by Dense U-Net structure based on intra-slice information. Then an inter-slice attention module is applied to refine the rough segmentation results using inter-slice information and produce the final segmentation results.

### 3.1. Dense U-Net Structure

The typically-used architecture for medical image segmentation is encoder-decoder structure based on U-Net (Ronneberger et al., 2015). However, there may be information loss during convolution. Recent work (Huang et al., 2017) (Jegou et al., 2017) has shown that short connections between layers could merge the contextual information before and after the convolution layers, therefore make convolutional networks more accurate and efficient. In this work, we modify the structure of classic U-Net via appending dense connections to its convolutional layers. It is designed to gain better capacity of extracting features among different slices, which is of great importance for increasing the accuracy. For segmentation

of 3D images, Dense U-Net is utilized to obtain the predicted probability results based on intra-slice information of each slice.

The detailed structure of the convolution blocks of encoder (DownBlock) and decoder (UpBlock) of Dense U-Net is shown in Figure1. The DownBlock includes convolutional layers (Conv), batch normalization (BN), rectified linear units (ReLU), dense connections (DC) and max-pooling layers (Maxpooling) while the UpBlock includes concatenation from encoder, Conv, BN, ReLU, DC and de-convolution layers (Deconv) .

### 3.2. Inter-slice Attention Module

Using 2D Dense U-Net structure can produce segmentation results based on intra-slice contextual information. However, the spatial information of continuity between adjacent slices is neglected, which is a restriction of higher performance in 3D image segmentation tasks. Due to the spatial continuity, the segmentation results of each layer are spatially correlated with the upper and lower layers. Therefore, the information of adjacent slices is useful for the segmentation of each single slice of the image.

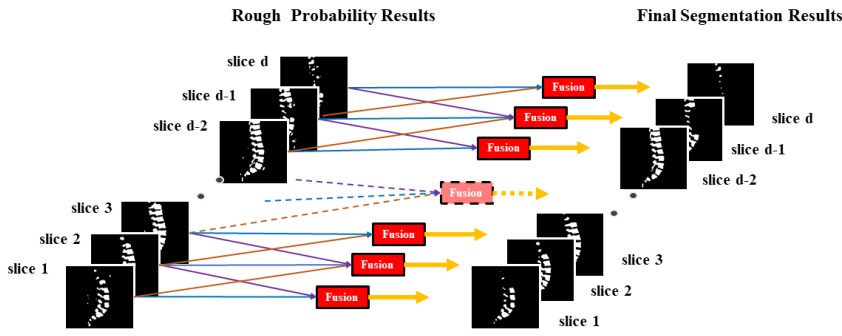

Figure 2: The overall structure of inter-slice attention module. For the refinement of segmentation results of each slice, the information of adjacent slices are used.

To address the issue, we propose an inter-slice attention module (ISA) to utilize contextual information between adjacent slices and augment the continuity of segmentation results. For segmentation tasks, attention is usually achieved by creating masks that represent an informative region on feature maps, so as to highlight the most salient regions and suppress irrelevant regions.

The overall structure of the inter-slice attention module is shown in Figure2. In order to utilize the information of spatial continuity, the feature maps of adjacent slices are used for the segmentation by generating attention masks and fusing into the feature maps of the slice, therefore the refined segmentation results are obtained.

The detailed structure of attention fusion module is shown in Figure3. For the segmentation of slice i, the feature maps of slice i+1 and/or (the first and last slice only have one adjacent slice) slice i-1 are utilized by generating attention masks using $1 \times 1$ convolutional layers and sigmoid. The generated masks represent the attention area for the segmentation

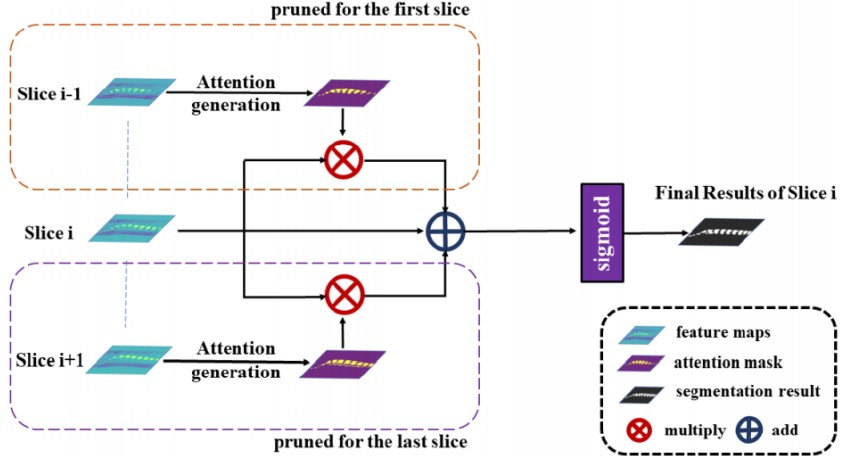

Figure 3: The detailed structure of the proposed attention fusion module.

of slice i based on information of adjacent slices. Then the masks are fused with feature maps of slice i. The size of the masks is the same as which of the feature maps of slice i. By multiplying the corresponding elements and adding the feature maps to the results after multiplication, we get the attention fusion output. For the first and the last slice, the process of attention fusion is halved. At the end of the module, sigmoid activation function is applied to get the final segmentation results.

## 4. Experimental results

### 4.1. Datasets and Evaluation Metrics

We use SpineSegT2W dataset to train and validate the models. The dataset contains 195 T2-weighted spine MR images of patients with disc herniation and degenerative. Each image is paired with ground truth labeled by expert radiologists. In the direction of the sagittal viewpoint, the size of each slice varies from $512 \times 512$ to $960 \times 960$ and the number of slices varies from 12 to 15 for different patients. Besides, we applied SAU-Net to an openly available MRI dataset in (Chu et al., 2015) for testing. The dataset contains 23 T2-weighted turbo spin echo MR images from 23 patients and the associated ground truth segmentation. The size of sagittal slices is $305 \times 305$ and the number of slices is 39 for all MR Images.

The Dice similarity coefficient, Jaccard coefficient, Hausdorff Distance (HSD), precision and recall were used to evaluate our segmentation results.

### 4.2. Implementation Details

Our following experiments are implemented using Keras with TensorFlow backend on NVIDIA Tesla V100 GPUs (32GB memory) . All networks use binary cross-entropy as loss function and Adam optimizer with initial learning rate 0.0001 for the training. Experimental results are validated by random 5-fold cross-validation by randomly shuffling the sequence of im-

ages and splitting the dataset into 5 fixed folds with 39 MR images in each fold, using 4 folds for training and the other one for testing. Therefore, we could achieve a more comprehensive evaluation of these models by analysing these results. The average training time of SAU-Net was approximate 5 hours on a single standard NVIDIA Tesla V100 GPU.

## 4.3. Comparison Experiments

In the conparison experiments, we compare SAU-Net with 2D segmentation methods like U-Net (Ronneberger et al., 2015) and 3D segmentation methods like 3D U-Net (Cicek et al., 2016) on the same experimental environment. The overall cross validation results of the comparison experiments are shown in Table4.3.

Table 1: Cross validation results of comparison experiments on SpineSegT2W dataset

| Models | fDSC (%) | Precision (%) | Recall (%) | Jaccard (%) | HSD (mm) |
|--------|----------|---------------|------------|-------------|----------|
| 2D U-Net | $88.06 \pm 0.16$ | $86.43 \pm 1.24$ | $89.85 \pm 0.84$ | $78.86 \pm 0.62$ | $3.12 \pm 0.035$ |
| 3D U-Net | $89.28 \pm 0.33$ | $89.35 \pm 0.35$ | $89.54 \pm 0.48$ | $80.72 \pm 0.88$ | $1.94 \pm 0.026$ |
| **SAU-Net** | $\mathbf{89.86 \pm 0.10}$ | $\mathbf{89.75 \pm 0.24}$ | $\mathbf{90.41 \pm 0.36}$ | $\mathbf{81.56 \pm 0.20}$ | $\mathbf{1.76 \pm 0.023}$ |

Table 2: Comparison of time consumption and model size of different methods

| Methods | Time Consumption | Model Size |
|---------|------------------|------------|
| 2D U-Net | 153s | 35M |
| 3D U-Net | 534s | 83M |
| SAU-Net | 178s | 38M |

According to Table4.3, SAU-Net achieve better segmentation results on all metrics compared with the other competitive models. As shown in Table2, comparing with using 3D convolutions, SAU-Net could achieve more accurate and robust segmentation with less cost of time consumption. We could get the conclusion that SAU-Net collaborate inter-slice information and intra-slice information collaborate in a better way and outperform other architectures on spine segmentation tasks.

To validate the effectiveness and robustness of our method, we also conduct experiments on the lower spine MRI dataset in (Chu et al., 2015). We evaluated our method on the dataset by running leave-one-out experiments as the way they used in (Chu et al., 2015). We can see that our method achieved better performance on all metrics. Additionally, we tested the well-trained model from SpineSegT2W dataset on the lower spine MRI dataset. As shown in Table3, the result indicates the effectiveness and good generalization capability of our method.

Table 3: Comparison of spine segmentation results on lower spine MRI dataset

| Models | DSC (%) | Precision (%) | Recall (%) | Jaccard (%) | HSD (mm) |
|---|---|---|---|---|---|
| SAU-Net (pre-trained) | 89.31 | 88.33 | 90.56 | 80.80 | 4.29 |
| SAU-Net* | **91.3 ± 0.5** | **95.1 ± 0.6** | **87.7 ± 1.3** | **83.8 ± 0.7** | **3.0 ± 0.8** |
| Chu et al* | 88.7 ± 2.9 | - | - | - | 6.4 ± 1.2 |

Note: ⋆ denotes leave-one-out experimental results; - denotes the result is not reported;

### 4.4. Ablation experiments

To demonstrate the effectiveness of inter-slice information extraction of SAU-Net, we performe a comparative performance experiment for spine segmentation with and without the inter-slice attention module (ISA) . The performance represented in Table4 demonstrated that the addition of ISA could refine the segmentation results and therefore get better results on all metrics.

Table 4: Cross validation results of ablation experiments SpineSegT2W dataset

| | Dice (%) | Precision (%) | Recall (%) | Jaccard (%) | HSD (mm) |
|---|---|---|---|---|---|
| Ours w/o ISA | 88.89 ± 0.20 | 87.83 ± 0.94 | 90.34 ± 0.86 | 80.11 ± 0.50 | 2.87 ± 0.039 |
| **Ours w/ ISA** | **89.86 ± 0.10** | **89.75 ± 0.24** | **90.41 ± 0.36** | **81.56 ± 0.20** | **1.76 ± 0.023** |

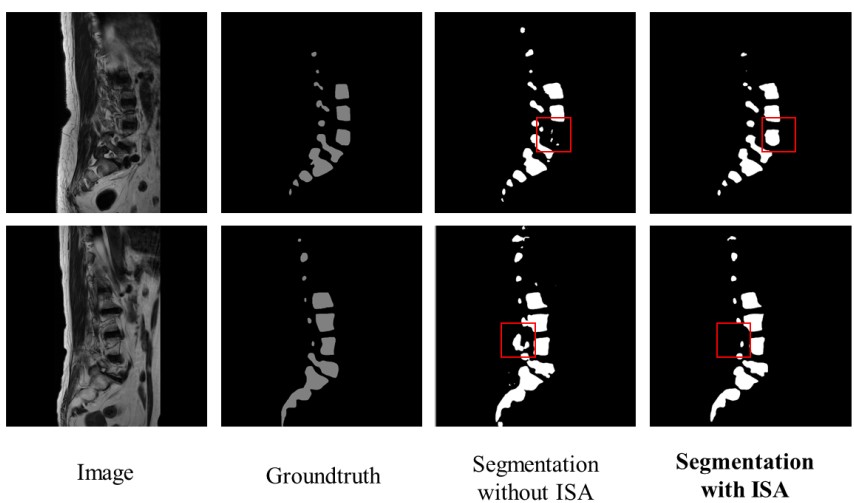

| Image | Groundtruth | Segmentation without ISA | **Segmentation with ISA** |

Figure 4: Examples of spine segmentation results with and without ISA.

Although the improvement of ISA in dice coefficient is not significant, according to the visualization of segmentation results in Figure4, we can see that the segmentation results of some areas that tend to be misclassified by 2D segmentation method are significantly

improved with the application of ISA. As a supporting evidence, the performance on Hausdorff distance are significantly improved. Therefore, the improvement of ISA could improve the accuracy and practicability of segmentation results without excessive computation cost, so as to assist in clinical treatment.

## 5. Conclusion

In this paper, we apply the attention mechanism to the utilization of inter-slice information on 3D image segmentation tasks based on 2D convolutional networks and proposed a novel structure called spatial attention-based densely connected U-Net (SAU-Net) for spine segmentation on 3D MR images. The architecture address the problems that 2D convolutions ignore the spatial information between adjacent slices and 3D convolutions suffer from high computation and memory cost, and risk of overfitting. Our method could substantially improve the segmentation accuracy and efficiency by fusing intra-slice and inter-slice features, which is crucial in the clinical practice. In addition, the strategy of using attention mechanism for extraction of inter-slice information could be easily adopted to other 3D image segmentation problems. Our future work includes more comprehensive validation and improvement of ISA and SAU-Net as well as possible application prospects for other tasks like multi-class segmentation and detection based on 3D medical images.

## Acknowledgments

This work is supported by the National Key Research and Development Program of China under Grant 2016YFF0201002, the University Synergy Innovation Program of Anhui Province (Grant Number: GXXT-2019-044), and the National Natural Science Foundation of China under Grant 61301005.

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
