# OpenReview forum: "SAU-Net: Efficient 3D Spine MRI Segmentation Using Inter-Slice Attention"
_MIDL.io/2020/Conference — MIDL 2020_

### Official Review · AnonReviewer3 · 2020-03-09
**2D U-net-like network with second step that incorporates information from neighboring slices**

**Rating:** 2
**Confidence:** 4

**Summary:**

This paper presents a method for segmentation of the vertebrae in lumbar spine MRI scans. The method is based on a dense 2D U-net and an additional refinement step based on an attention mechanism. This refinement step takes information from the previous and the following slice into account to refine the 2D segmentation mask. The method was evaluated with cross validation on a set of approximately 200 MRI scans and compared with a 2D and a 3D U-net.

**Strengths:**

- Vertebra segmentation in MRI is a challenging task, many previous publications focused on segmentation of only the vertebral bodies
- The presented method is not very complex
- The method is evaluated on a sizable dataset

**Weaknesses:**

- The introduction mentions several vertebra segmentation methods, but in the experiments the authors compare their method not with any of these state-of-the-art methods but with a standard 2D and 3D U-net
- The improvement in segmentation performance even with these non-optimal baselines is rather small
- There is no evaluation on a public dataset, such as https://doi.org/10.5281/zenodo.22304
- The description is not always clear, for example, it is not clear what the "stacked" part of the "stacked dense U-net" is

**Detailed Comments:**

- The MRI slices are upside down in all figures (with the sacrum at the top)
- The residual connections in Figure 1 are ambiguous, e.g., there is an arrow pointing to the first "plus" in the UpBlock, but to my understanding there should also be an arrow coming from there to the second "plus". It might be better to use individual arrows / lines for each individual connection. The same applies to the DownBlock.
- Figure 2 is very helpful
- Reporting both Dice and Jaccard scores is not very useful, and using precision and recall for evaluation of a segmentation task is also not very informative - a surface distance measure such as the Hausdorff distance would be more helpful
- The number of scans in the dataset is unclear, the text mentions "... contains 195 and 20 sagittal ..." - but the 5-fold cross-validation mentions 39 * 5 = 195 scans, so what happened with the other 20 scans?
- It would be useful to explicitly mention that the 2D slices in this application are sagittal and not axial slices, which I think can be assumed to be the default understanding of "slice"

**Justification Of Rating:**

While this is a very relevant application and overall a sound method that combines a 2D segmentation network with a mechanism for incorporating information from neighboring slices, the evaluation of this method needs to be improved. A comparison with a state-of-the-art method is missing and the used metrics are not ideal.

**Paper Type:**

methodological development

**Special Issue:**

no

---

> ### Author Response · Authors · 2020-03-25
> **Response to AnonReviewer3**
>
>     We sincerely appreciate the comments provided by the reviewer, which will undoubtedly contribute to the quality of the revised manuscript if it is finally accepted. Please find detailed, point-by-point responses to raised clarification requests below.
>
> 1. The experiments the authors compare their method only with a standard 2D and 3D U-net
>     The main purpose of this work is to propose a novel 2.5D method for 3D medical image segmentation problem that could segment accurately with moderate computational cost. Since the model is modified from U-Net, validation among our method with U-Net and 3D U-Net is the most straight forward way to prove the effectiveness of our modification. Moreover, 2D and 3D U-Net have been validated on many MR image segmentation problems. Their performances demonstrate their superiority for MR image segmentation. Hence, the comparison could strongly support the effectiveness of our method.
>
> 2. Comments on valuation metrics.
>     “A surface distance measure such as the Hausdorff distance would be more helpful.” We agree with the reviewer and will use Hausdorff distance as an additional metrics in our experiments and discuss the corresponding results in the final version of the manuscript. Some results are as follows: Hausdorff distance for 2D U-Net,3D U-Net and SAU-Net in 5-fold cross-validation : 3.12±0.035,1.94±0.026,1.76±0.023
>
> 3. The improvement of segmentation performance is rather small.
>     Sure, compared with 3D U-Net, the performance gain is not very big. However, the computational cost of the proposed method is significantly lower than 3D U-Net. As for this problem, we have discussed it in Section 4.4. Though the improvement of Dice score may be not very significant, in Figure 4 we could visualize that with the application of ISA, the segmentation results of some areas that tend to be misclassified by 2D segmentation method are significantly improved. As a supporting evidence, the improvement is more significant on Hausdorff distance. Compared with 3D U-Net, we could achieve comparative segmentation accuracy (even slightly higher) with less computation cost. Therefore, we do think it is a useful application in 3D medical image segmentation.
>
> 4. There is no evaluation on a public dataset
>     We thank the reviewer for pointing out this issue and providing this public dataset (https://doi.org/10.5281/zenodo.22304) . We have supplemented the evaluation on this dataset and will report them in the final version of the manuscript. Some preliminary results are as follows: test result with pre-trained SAU-Net Dice:89.12, Hausdorff distance:4.29. Leave-one-out experimental results of SAU-Net Dice:91.3±0.5, Hausdorff distance:3.0±0.8. Compare with results of the dataset paper：Dice:88.7±2.9, Hausdorff distance:6.4±1.2
>
> 5. It is not clear what the "stacked" part of the "stacked dense U-net" is
>     As for stacked Dense U-Net, we intended to express that the overall segmentation and evaluation is stacked slices-based, instead of single slice-based in 2D U-Net. We rephrased this sentence by “In this work, we modify the structure of classic U-Net via appending dense connections to its convolutional layers. It is designed to gain better capacity of extracting features among different slices, which is of great importance for increasing the accuracy. For segmentation of 3D images, dense U-Net is utilized to obtain the predicted probability volumes based on intra-slice information of each slice.” We will revise our manuscript to make the description clearer.
>
> 6. The ambiguousness of figures
>     We sincerely appreciate these useful comments and will revise them in our final version.
>
> 7. The number of scans in the dataset
>     The dataset contains 215 images, where 195 for training and 20 for testing. The 195 scans are released for training while the other 20 scans are reserved for testing and the labels are not available. The overall experiments and 5-fold CV in this paper were performed on these 195 scans. We will reorganize the paper for clearer description of the dataset. Besides, experiments on provided public dataset (https://doi.org/10.5281/zenodo.22304) will be supplemented in our final version.
>
> 8. It would be useful to explicitly mention that the 2D slices in this application are sagittal and not axial slices.
>     We sincerely appreciate this useful comment. The default description will be supplemented in our final version.
>
>     Once again, we thank the reviewer for his / her useful insights, which have been of great help.

---

### Official Review · AnonReviewer1 · 2020-03-11
**Review of SAU-Net (54)**

**Rating:** 3
**Confidence:** 5
**Recommendation:** Poster

**Summary:**

The paper describes a new segmentation method to segment vertebrae in spine MRIs. The method was validated on the SpinesegT2W dataset (seems to be an internal closed source dataset) with 190 sagittal T2-Weighted MRIs. For comparison the paper compared the performance of the proposed method against several other methods namely the original U-Net and 3D U-Net.

**Strengths:**

- The idea of using a 2D CNN coupled with an inter-slice refinement step to help 3D segmentation is novel and interesting
- The ISA bit of the paper should be easily applicable to other segmentation methods working on 3D volumes
- Better performance than 3D U-Net
- Validated on a large dataset

**Weaknesses:**

- The structure of the paper can be better
- Some bits of the paper can be a bit clearer


Detailed comments:
- Section 3.1. The following sentence is a bit unclear "For better extract of intra-slice feature, we redesign the structure of convolution blocks of classic U-Net, proposed stacked Dense U-Net structure for rough segmentation based on intra-slice information.".
- Section 3.2. "For segmentation tasks, attention is usually achieved by creating masks that represent an informative region on feature maps, so as to highlight the most salient regions and suppress irrelevant regions.". How are masks generated? Are they  thresholded or are they the raw output slices from SAU-Net?
- Section 4.1. "The dataset contains 195 and 20 sagittal T2-weighted spine MR images of patients ...". What does 195 and 20 here refer to?
- Section 4.4. it would be interesting to see the effects of using ISA on the 2D U-Net and 3D U-Net.
- Section 2.2. Missing space between features and for in the sentences "... the most discriminant features.For medical image segmentation"
- Section 3. Figure1 -> Figure 1. Fix throughout paper.
- Acknowledgements should not be a numbered section.

**Justification Of Rating:**

The paper presents a method to easily improve 3D segmentation that should be easily applicable to other methods. The proposed method also presented good performance on multiple metrics compared to other segmentation methods. The paper is well validated but there are minor weaknesses that need to be addressed.

**Paper Type:**

methodological development

**Special Issue:**

no

---

> ### Author Response · Authors · 2020-03-25
> **Response to AnonReviewer1**
>
>     We sincerely appreciate the comments provided by the reviewer, which will undoubtedly contribute to the quality of the revised manuscript if it is finally accepted. We thank the reviewer for his/her time and positive assessment of our work. Please find detailed, point-by-point responses to raised clarification requests below.
>
> 1. Some sentences in Section 3.1 is a bit unclear.
>     We rephrased this sentence by “In this work, we modify the structure of classic U-Net via appending dense connections to its convolutional layers . It is designed to gain better capacity of extracting features among different slices, which is of great importance for increasing the accuracy. For segmentation of 3D images, Dense U-Net is utilized to obtain the predicted probability volumes based on intra-slice information of each slice.” As for stacked Dense U-Net, we intended to express that the overall segmentation and evaluation is stacked slices-based, instead of single slice-based in 2D U-Net. We will revise our manuscript to make the description clearer.
>
> 2. Section 3.2 How are masks generated?
>     The attention generation in 3.2 use 1x1 conv + sigmoid to create masks for previous/following slice. Then the generated masks are fused with the feature maps of the slice at hand to create the final segmentation results. We will reorganize the manuscript to make the description clearer.
>
> 3. What does 195 and 20 here refer to?
> The dataset contains 215 images, where 195 for training and 20 for testing. The 195 scans are released for training while the other 20 scans are reserved for testing and the labels are not available. The overall experiments and 5-fold CV in this paper were performed on these 195 scans. We will reorganize the paper for clearer description of the dataset. Besides, thanks to reviewer3, we have supplemented the evaluation on experiments on another public dataset (https://doi.org/10.5281/zenodo.22304) and will report them in the final version of the manuscript.
>
> 4. Comments on the format of the article.
>     We sincerely appreciate these useful comments and will revise them in our final version.
>
>     Once again, we thank the reviewer for his / her useful insights, which have been of great help.

---

### Official Review · AnonReviewer4 · 2020-03-12
**A 2.5D approach for MRI spine segmentation with not completely clear motivation and somewhat surprising results**

**Rating:** 3
**Confidence:** 4
**Recommendation:** Poster

**Summary:**

A 2D U-NET based approach in combination with adjacent slice fusion is proposed for MRI spine segmentation. The method is compared to a pure 2D U-NET and a 3D U-Net, with the proposed method outperforming both when evaluated on the Spineseg T2W data set. A DICE score, precision and recall all of about 90% is reached with the method.

**Strengths:**

The paper has the clear proposition to make use of the 3D context for spine segmentation while avoiding the runtime performance cost of a 3D U-NET. The approach is evaluated on a public data set and compared to 2 alternative approaches (2D and 3D U-NET).

**Weaknesses:**

The motivation is not completely clear. The authors state quite generally, that 3D CNNs suffer from high computational and memory costs without relating it to respective boundary conditions for a given clinical workflow or to known problems.
Later on the computational and memory costs are not detailed out for the 3 approaches, it is only generally stated that their approach is 3 times faster than the 3D U-Net approach.

The concept of the attention mechanism is not completely clear. It is also somewhat misleading, since the attention mechanism is a post-processing step, fusing the 2D results of the slice at hand with the result of the previous and following slice. It is not, as one could assume, a mechanism that steers a neural network in a way giving more weight to an attention area or the like. The attention mechanism is not really described in detail and seems to be some kind of a slice averaging appoach. What exactly is the 'attention generation' step in fig. 3?

**Justification Of Rating:**

The paper is interesting and sound enough to be accepted, but there are too many unclarities and weak points to justify a strong accept. Especially the attention module needs to be explained in more detail and the authors should reason why their approach outperforms a 3D U-NET.

**Paper Type:**

validation/application paper

**Questions To Address In The Rebuttal:**

-How exactly works the attention mechanism?
-Why does the proposed method outperform the 3D U-NET? It seems that the attention based slice fusion is below the capabilities of a 3D U-NET.

**Special Issue:**

no

---

> ### Author Response · Authors · 2020-03-25
> **Response to AnonReviewer4**
>
>     We sincerely appreciate the comments provided by the reviewer, which will undoubtedly contribute to the quality of the revised manuscript if it is finally accepted. Please find detailed, point-by-point responses to raised clarification requests below.
>
> 1. The motivation is not completely clear. The authors state quite generally, that 3D CNNs suffer from high computational and memory costs without relating it to respective boundary conditions for a given clinical workflow or to known problems.
>     Technically, we aim at developing a CNN model to implement 3D segmentation with moderate computational and memory cost. Existing 3D methods are too redundant to meet the requirement of medical application.
>     Clinically, accurately delineate the boundary of vertebrae is helpful for automatic measurement, computer aided diagnosis and morphology-based research. The accuracy of segmentation directly affects 3D reconstruction and visualization, which is an essential step for lesion detection and diagnosis. We will reorganize the manuscript for clearer description illustration the motivation and important clinical significance.
>
> 2. Computational and memory costs are not detailed out.
>     Thank you for your advice, we will report the result in the revised manuscript. Some preliminary results are as follows: time consumption in the testing stage and size of model for U-Net, 3D U-Net and SAU-Net: 153s/35M, 534s/83M and 178s/38M respectively under same circumstances.
>
> 3. The concept of the attention mechanism is not completely clear. How exactly works the attention mechanism?
>     The attention mechanism is inspired by human attention that help people focus on key information. In image processing, the attention mechanism allows CNN to focus by keeping useful information and suppress these irrelevant parts. The common practice is to learn a probability mask, where the weights at informative parts are close to 1 and irrelevant part are nearly 0.
>     In this work, the “rough segmentation” are extracted feature maps instead of binary segmentation results. For inter-slice attention module (ISA), attention generation use 1x1 conv + sigmoid to create masks for previous/following slices and then fuse them with the feature maps of the slice at hand to create the final segmentation results. The application of ISA could steer the network to focus more on attention area instead of simply slice averaging. From Figure 4, we can see that the segmentation results of some areas that tend to be misclassified by 2D segmentation method are significantly improved by ISA.
>     We realized that some statement in the manuscript may be ambiguous, and we will reorganize the paper to make the description clearer for the final version.
>
> 4. Why does the proposed method outperform the 3D U-NET?
>     For 3D U-Net, the large number of parameters may result in higher risk of overfitting.  Additionally, the information loss during 2x2x2 maxpooling in 3D U-Net may reduce the segmentation performance, especially for 3D images with thicker slices. Our proposed method could utilize inter-slice information more effectively. As a result, from the cross validation results we observed that SAU-Net slightly outperform the 3D U-Net.
>
>     Once again, we thank the reviewer for his / her useful insights, which have been of great help.

---

> > ### Comment · AnonReviewer4 · 2020-04-03
> > **Not convinced**
> >
> > Dear authors thank you for comments.
> > To your comment 1.:
> > My point was specific clinical requirements. If you state that the current approaches are too slow, it would be helpful if you would state which runtime performance would be sufficient, same with memory consumption. You can do everything faster and better, the question is, what exactly do you need to achieve for a given problem or workflow.
> >
> > To your comment 2.:
> > thank you
> >
> > To your comment 3.:
> > Still not clear to me. An attention mechanism, also as you describe it, would stear the attention of a neural network or some other algorithm which is sufficiently complicated for the notion of 'attention'. I have the impression that in your approach, it is a straight forward fusion of neighboring results.
> >
> > To your comment 4.:
> > Not completely convinced. Are there signs of overfitting? Was it addressed by regularization? Why should your approach be better suited to use information from neighboring slices? What would a 3D U-net not be able to do what your approach is doing? I still think there is the danger of using the 3D Unet in a non-optimal setting for the task

---

> > > ### Author Response · Authors · 2020-04-05
> > > **Response**
> > >
> > > Thank you for your feedback.
> > >
> > > Response to comment 1:
> > >
> > >      Thank you for your question. To meet the clinical requirement, ideally, 100% accuracy is the best, but it is impossible to achieve. The specific clinical requirement is difficult to estimate, since the requirement varies form different diseases, doctors’ diagnostic habits, and how AI contribute to the diagnostic workflow. In addition, where the segmentation error occurs, how the error will negatively influence the diagnosis, are also important, and cannot easily quantified.
> > >      However, considering that the AI system for spine segmentation is still not widely used for computer aided diagnosis, and some prototype system still require human correction before application. We could conclude that existing methods still do not meet the clinical requirement. Hence, our improvement on the performance is worthy.
> > >      As for the memory consumption, it would be better if it is less than 4GB RAM. Because, in some small hospitals in developing country, it may be difficult for them to afford an advanced computer. For 3D U-Net, the memory consumption is far beyond. Therefore, 2.5D CNN methods with less consumption cost and comparative accuracy could be valuable solutions to this problem.
> > >
> > > Response to comment 3:
> > >
> > >      Our proposed inter-slice attention module is designed to steer the network to focus on regions where adjacent slices are probably vertebrae (based on intra-slice information only). These regions of slice at hand may have higher probability to be vertebrae due to spatial continuity.
> > >      Though unlike common usage of attention, our multiply/attention fusion process are based on extracted feature maps instead of input images. Because, shape differences of adjacent slices still exist, direct fusion of adjacent prediction may influence the intra-slice information extraction. Besides, the design of ISA module is trainable, instead of a straight forward fusion process.
> > >
> > > Response to comment 4:
> > >
> > >      Yes, there are signs of over fitting. For 3D U-Net, the Dice performance on training set is nearly 0.94 but 0.89 for testing set. In our design the performance of SAU-Net is 0.91 and 0.89, respectively, where the overfitting is not obvious. For the 3D U-Net, we tried the regularization but still not capable to increase the performance on testing set.
> > >      Besides, 3D networks may not always the best option. As [Abulnaga et al 2018] (https://arxiv.org/abs/1811.01085) concluded, “The 3D networks performed poorly. We observed that their increased number of parameters resulted in more overfitting. Additionally, they were unable to take full advantage of the third image dimension, due to the large number of scans with only 2 axial slices.” Additionally, the memory consumption is also large, which hinders its application in real clinical application.

---

### Meta-Review · Area_Chair1 · 2020-04-07
**MetaReview of Paper54 by AreaChair1**

**Rating:** 3
**Recommendation For Accepted Papers:** Poster

**Metareview:**

The majority of reviews are generally positive. The good performance compared to a baseline 2D and 3D U-Net implementation is promising. The authors have addressed the major concerns raised by the negative review in their rebuttal sufficiently well. The requested surface-based evaluation metrics show again a promising performance of the proposed method and can be added to the final paper.

**Paper Type:**

methodological development

**Special Issue:**

no

---

### Decision · Program_Chairs · 2020-04-11

Accept